# DNA polymerase V activity is autoregulated by a novel intrinsic DNA-dependent ATPase

Aysen L Erdem[1†], Malgorzata Jaszczur[1†], Jeffrey G Bertram[1], Roger Woodgate[3], Michael M Cox[4], Myron F Goodman[1,2]*

[1]Department of Biological Sciences, University of Southern California, Los Angeles, United States; [2]Department of Chemistry, University of Southern California, Los Angeles, United States; [3]Laboratory of Genomic Integrity, National Institute of Child Health and Human Development, National Institutes of Health, Bethesda, United States; [4]Department of Biochemistry, University of Wisconsin–Madison, Madison, United States

**Abstract** *Escherichia coli* DNA polymerase V (pol V), a heterotrimeric complex composed of UmuD'$_2$C, is marginally active. ATP and RecA play essential roles in the activation of pol V for DNA synthesis including translesion synthesis (TLS). We have established three features of the roles of ATP and RecA. (1) RecA-activated DNA polymerase V (pol V Mut), is a DNA-dependent ATPase; (2) bound ATP is required for DNA synthesis; (3) pol V Mut function is regulated by ATP, with ATP required to bind primer/template (p/t) DNA and ATP hydrolysis triggering dissociation from the DNA. Pol V Mut formed with an ATPase-deficient RecA E38K/K72R mutant hydrolyzes ATP rapidly, establishing the DNA-dependent ATPase as an intrinsic property of pol V Mut distinct from the ATP hydrolytic activity of RecA when bound to single-stranded (ss)DNA as a nucleoprotein filament (RecA*). No similar ATPase activity or autoregulatory mechanism has previously been found for a DNA polymerase.

*For correspondence:
mgoodman@usc.edu

†These authors contributed equally to this work

**Competing interests:** The authors declare that no competing interests exist.

**Reviewing editor**: Michael R Botchan, University of California, Berkeley, United States

## Introduction

DNA polymerase V is a low fidelity TLS DNA pol with the capacity to synthesize DNA on a damaged DNA template (*Reuven et al., 1999*; *Tang et al., 1999*). It is encoded by the LexA-regulated *umuDC* operon and is induced late in the SOS response in an effort to restart DNA replication in cells with heavily damaged genomes (*Goodman, 2002*). The enzyme is responsible for most of the genomic mutagenesis that classically accompanies the SOS response (*Friedberg et al., 2006*).

RecA protein plays a complex role in the induction of pol V. As a filament formed on DNA (the form sometimes referred to as RecA*), RecA* facilitates the autocatalytic cleavage of the LexA repressor (*Little, 1984*; *Luo et al., 2001*). This leads directly to the induction of the SOS response. Some 45 min after SOS induction, those same RecA* filaments similarly facilitate the autocatalytic cleavage of UmuD$_2$ protein to its shorter but mutagenically active form UmuD'$_2$ (*Burckhardt et al., 1988*; *Nohmi et al., 1988*; *Shinagawa et al., 1988*). UmuD'$_2$ then interacts with UmuC to form a stable UmuD'$_2$C heterotrimeric complex (*Woodgate et al., 1989*; *Bruck et al., 1996*; *Goodman, 2002*). UmuD'$_2$C copies undamaged DNA and performs TLS in the absence of any other *E. coli* pol (*Tang et al., 1998*), but only when RecA* is present in the reaction. UmuD'$_2$C (*Tang et al., 1998*; *Karata et al., 2012*) or UmuC (*Reuven et al., 1999*), have minimal pol activity in the absence of RecA*. Final conversion of the UmuD'$_2$C complex to a highly active TLS enzyme requires the transfer of a RecA subunit from the 3′ end of the RecA* filament to form UmuD'$_2$C-RecA-ATP, which we refer to as pol V Mut (*Jiang et al., 2009*).

**eLife digest** DNA polymerases are enzymes that copy the genetic material within a cell using a strand of an existing double helix as a template to guide the synthesis of a new DNA strand. Since DNA is continuously exposed to damaging agents, and damaged DNA can derail the DNA replication machinery; a cell must either repair or bypass (via a process called translesion synthesis) this damage to copy its genome.

Most living things, from bacteria to humans, have specific DNA polymerases for translesion synthesis that can copy past damaged DNA, such as DNA Polymerase V in the bacterium *E. coli*. However, DNA polymerase V will also often introduce mistakes when it copies DNA that is not damaged—and cells will subsequently switch to use a different polymerase to accurately copy undamaged DNA.

DNA polymerase V is activated by binding to a protein called RecA and a molecule of adenosine triphosphate (ATP for short). ATP stores energy, which cells release by breaking down the molecule into simpler chemicals: but how do ATP and RecA work together to activate this polymerase?

Now, Erdem, Jaszczur et al. have addressed this question using biochemical techniques on purified polymerases, proteins and DNA fragments in a test tube. These experiments reveal that DNA polymerase V must bind to an ATP molecule before it can attach to the DNA template, and must remain bound to ATP while synthesizing the new DNA strand. After the activated polymerase has attached to the DNA, it will break down the molecule of ATP to free itself from the DNA. Furthermore, although the RecA protein can also break down ATP, Erdem, Jaszczur et al. found that a mutant RecA without this ability could still activate DNA polymerase V to break down this molecule itself.

Binding to and breaking down a molecule of ATP by a DNA polymerase has not been observed before as a method of directly regulating these enzymes' activity. Erdem, Jaszcuzur et al. suggest that, in living cells, this extra level of control would limit how long the DNA polymerase V spends attached to the DNA. As such, this polymerase would only be used to copy stretches of damaged DNA, but would not continue on to copy neighboring stretches of undamaged DNA where it would likely introduce new errors.

ATP plays an essential but heretofore enigmatic role in the activation process. Activation can proceed with ATP or the poorly-hydrolyzed analogue ATPγS. ATP is part of the active complex, with approximately one molecule of ATP per active enzyme (*Jiang et al., 2009*). Under some conditions, activated and isolated pol V Mut exhibited polymerase activity only when additional ATP or ATPγS was added to the reaction (*Jiang et al., 2009*). The function of the ATP complexed with pol V Mut is delineated in this report.

## Results

Throughout this study, we utilize three variants of RecA protein for pol V activation. One is the wild-type (WT) RecA protein, which activates moderately in solution. The second is the RecA E38K/ΔC17 double mutant. The E38K mutation results in faster and more persistent binding of RecA protein to DNA, and the deletion of 17 amino acid residues from the RecA C-terminus eliminates a flap that negatively autoregulates many RecA activities (*Lavery and Kowalczykowski, 1992*; *Eggler et al., 2003*). The combination results in a RecA protein that activates pol V much more readily in vitro (*Schlacher et al., 2006*; *Jiang et al., 2009*). The third RecA variant is RecA E38K/K72R, combining the E38K change with the K72R mutation that all but eliminates the RecA* ATPase activity (*Gruenig et al., 2008*).

### ATP activation of pol V Mut

Pol V Mut can be formed and effectively isolated by incubating UmuD′$_2$C complexes with RecA* that is bound to ssDNA oligonucleotides tethered to streptavidin-coated agarose beads, spinning the beads out of solution to remove RecA*, and taking the now active pol V Mut from the supernatant. In this initially described protocol (*Jiang et al., 2009*), a small amount of ATP or ATPγS is transferred adventitiously from the supernatant with the pol V Mut. When WT RecA protein is used in this activation, the isolated pol V Mut WT is active only if supplemental ATP or ATPγS is added to the reaction

mixtures (*Jiang et al., 2009*). However, when a RecA variant that provides more facile activation of pol V was used, RecA E38K/ΔC17, the added ATP or ATPγS was apparently not needed for pol V Mut function (*Jiang et al., 2009*). The reason for this disparity in the ATP requirement for pol V Mut function is resolved below.

To explore the role of ATP, an amended protocol (outlined in *Figure 1A*) was used that employed a spin column to more rigorously remove exogenous ATP or ATPγS after pol V Mut formation. As shown in *Figure 1B*, pol V Mut function now depends completely on added ATP or ATPγS when this activation protocol was utilized, regardless of which RecA variant was used in the activation. Pol V Mut is not activated by GTP, ADP or dTTP, (*Figure 1—figure supplement 1A*) and does not incorporate ATP or ATPγS into DNA during synthesis (*Figure 1—figure supplement 1B*). Thus, ATP or an ATP homolog is an absolute requirement for pol V Mut function. For pol V Mut WT, the addition of ATPγS supports synthesis, whereas ATP does not (*Figure 1B*). The same ATP effect was observed for pol V Mut WT synthesis on DNA containing an abasic site (*Figure 1—figure supplement 2*). dATP activation does not result in appreciable DNA extension (*Figure 1—figure supplement 1*). For pol V Mut E38K/K72R, synthesis is observed with either ATPγS, ATP or dATP (*Figure 1B*, *Figure 1—figure supplement 1*). Pol V Mut E38K/ΔC17 can also use ATP, ATPγS or dATP as a required nucleotide cofactor (*Figure 1B*, *Figure 1—figure supplement 1*). Notably, the requirement for ATPγS/ATP was completely masked in earlier studies of transactivation of pol V by RecA* filaments that remain in the solution with pol V Mut, because the ATPγS or ATP needed to form RecA* is always present in the transactivation reaction (*Schlacher et al., 2006*).

## Pol V Mut is a DNA-dependent ATPase

Pol V Mut does not simply require ATP or ATPγS for activity; it possesses an intrinsic DNA-dependent ATPase activity (*Figure 2*). This is unprecedented for a DNA polymerase. A very sensitive ATPase assay, based on the fluorescence of a $P_i$ binding protein, is used in this work. The assay permits observation of the first 5 μM of ATP hydrolyzed, which is limited by the concentration of the fluorescent $P_i$ binding protein found in solution. A 30 nt ssDNA oligomer (1 μM) is present in all reactions. In *Figure 2A*, results are shown with pol V Mut made with WT RecA protein (pol V Mut WT). WT RecA protein alone exhibits limited stability on short oligonucleotides when present at sub-micromolar concentrations. Limited ATP hydrolysis by RecA WT alone is seen in this assay; the initial filaments produce a burst of ATP hydrolysis and then level off to a much slower rate, presumably reflecting filament binding, dissociation, and slow reassembly. Both phases of the reaction exhibit a dependence on RecA WT protein concentration (*Figure 2A*, *Figure 2—figure supplement 1A*). In contrast, after detectable free RecA protein and RecA* have been removed, equivalent concentrations of pol V Mut WT exhibit higher levels of ATP hydrolysis than do similar amounts of RecA alone (*Figure 2A*). The $P_i$ release rate constant ($k_{cat}$) in the presence of ssDNA, 12 nt and 3 nt over hang (oh) Hairpin (HP) and in the absence of DNA are summarized in *Table 1*. For the calculation of rates see 'Material and methods'.

In principle, the ATPase properties of pol V Mut might be determined mainly, if not solely, by the properties of its RecA subunit. But that's in fact not the case. To address whether the pol V Mut -associated ATPase activity can be distinguished from the intrinsic DNA-dependent ATPase activity of RecA, we assembled pol V Mut with a RecA (E38K/K72R) mutant deficient in ATPase activity (*Gruenig et al., 2008*). Using the same highly sensitive fluorescence-based assay as described for panel 2A, we were unable to detect DNA-dependent ATPase activity for RecA E38K/K72R above background (*Figure 2B*). However, in contrast, pol V Mut E38K/K72R exhibits substantial ATPase activity. The observed ATPase scales with the concentration of pol V Mut E38K/K72R (*Figure 2B*), as is also the case for pol V Mut WT (*Figure 2A*). Therefore, pol V Mut is a DNA-dependent ATPase (*Figure 2*). As RecA E38K/K72R protein hydrolyzes little or no ATP on its own (*Figure 2B*, *Figure 2—figure supplement 1B*), the pol V Mut activity cannot be attributed to a low level of contamination by RecA protein. A small amount of ATP hydrolysis above background can be observed with RecA E38K/K72R by increasing its concentration to 2.5 μM (84 nM Pi release/min, *Figure 2—figure supplement 1B*).

We previously reported that the active complex of pol V Mut is composed of pol V-RecA-ATP (*Jiang et al., 2009*). We can further distinguish pol V Mut from either pol V or free RecA in an assay measuring binding to etheno-ATP via anisotropy (*Figure 2C*). Pol V Mut made with either RecA WT or RecA E38K/K72R registers a substantial and concentration-dependent signal in this assay, while binding of etheno-ATP to either pol V or RecA alone is much weaker.

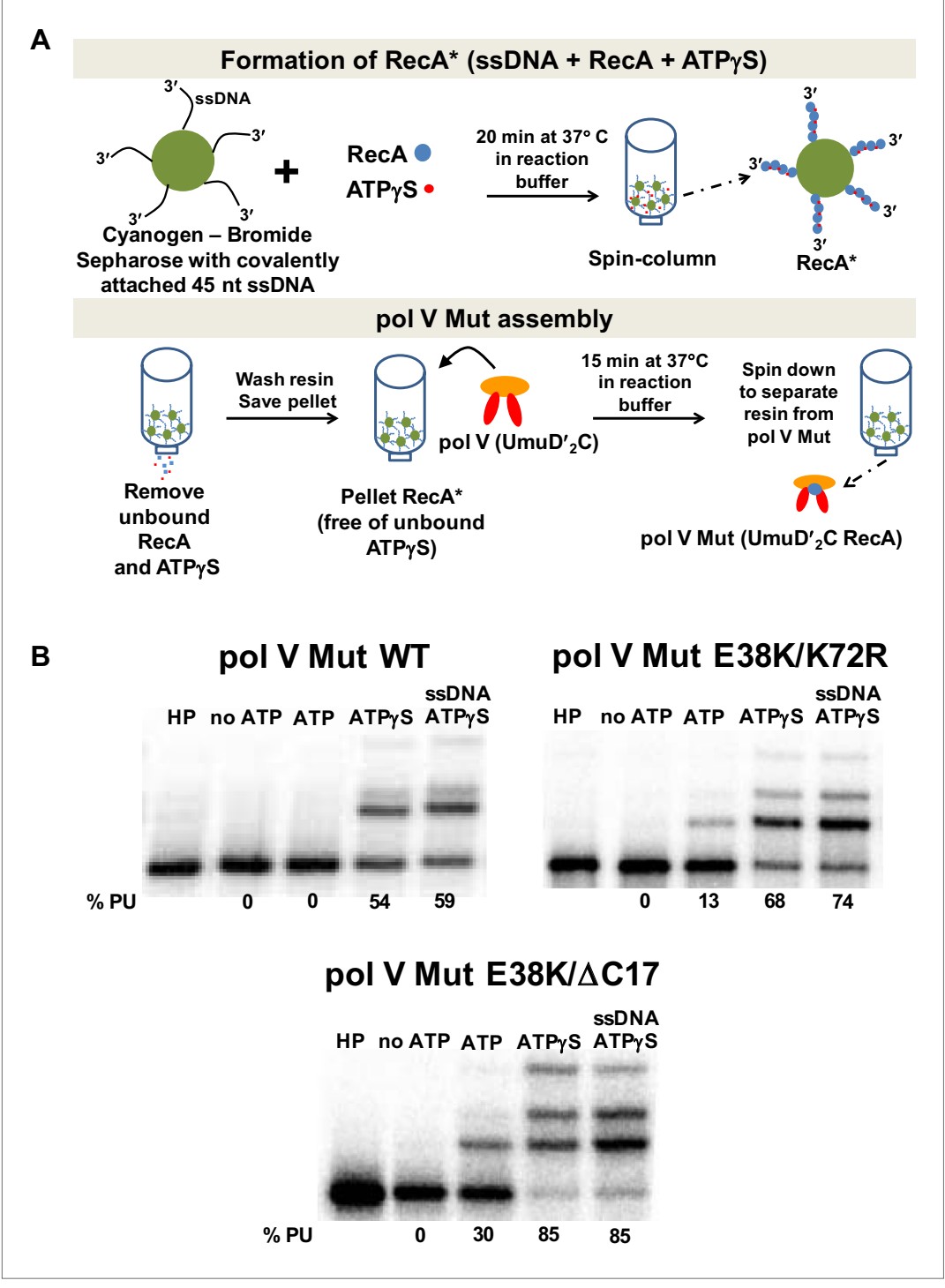

**Figure 1**. Pol V Mut requires ATP/ATPγS for activity. Sketch of pol V Mut assembly; pol V is activated by RecA* bound to Cyanogen-Bromide Sepharose resin as described in 'Materials and methods'. The pol V Mut assembly protocol ensures the separation of pol V Mut from free RecA, ssDNA and ATPγS. (**B**) Pol V Mut (400 nM) activity was detected on 5'-$^{32}$P-labeled 3 nt oh HP (100 nM) in the presence or absence of ATP/ATPγS and dNTPs. To detect free RecA in the pol V Mut solution, ssDNA and ATPγS was added to the reaction. Comparable activity levels between ATPγS alone and ATPγS + ssDNA indicate that pol V Mut is intact and free of RecA.

The following figure supplements are available for figure 1:

*Figure 1. Continued on next page*

*Figure 1. Continued*

**Figure supplement 1**. Pol V Mut is not activated by GTP, ADP, or dTTP and does not incorporate ATP/ATPγS in to DNA during synthesis.

**Figure supplement 2**. Pol V Mut WT activity on DNA containing an abasic site.

## Pol V Mut requires ATP/ATPγS to bind p/t DNA

To further explore the requirement for the addition of a ribonucleotide cofactor, we measured pol V Mut binding to p/t DNA and DNA synthesis as a function of the concentration of added ATPγS or ATP (*Figure 3*). Binding is absolutely dependent on a nucleotide cofactor. Pol V Mut WT binding to p/t DNA with a 3 nt template overhang (3 nt oh) increases roughly linearly up to about 180 µM ATPγS, saturating at about 220 µM (*Figure 3A*). However, binding is not detectable in the presence of ATP. Increasing the length of the template overhang to 12 nt has essentially no effect on binding as a function of nucleotide concentration (*Figure 3—figure supplement 1*). However, with ATPγS pol V Mut

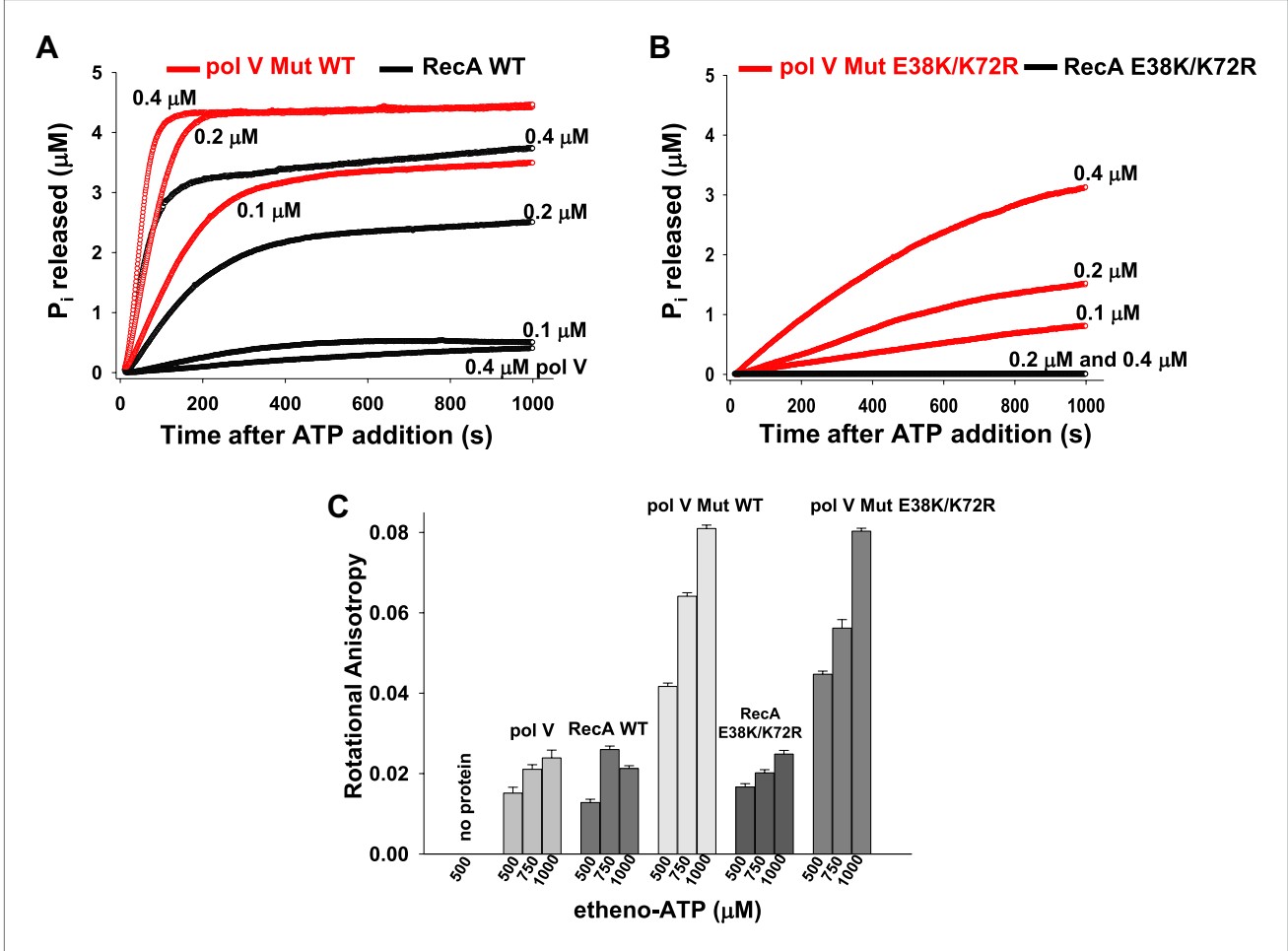

**Figure 2**. Pol V Mut is a DNA-dependent ATPase. (**A** and **B**) ATP hydrolysis by pol V Mut and RecA (0.1 µM, 0.2 µM and 0.4 µM each) was measured using MDCC-PBP (5 µM) in the presence of 30 nt ssDNA (1 µM) and ATP (500 µM). MDCC-PBP fluorescence increases as $P_i$ is released due to ATP hydrolysis. The measurements were taken at a resolution of 1 point per sec for approximately 1000 s. (**C**) Binding of 400 nM pol V, RecA and pol V Muts to etheno-ATP at varied concentrations was measured using rotational anisotropy. The error bars correspond to 1 SEM.

The following figure supplements are available for figure 2:

**Figure supplement 1**. RecA WT and RecA E38K/K72R-dependent ATP hydrolysis.

**Table 1.** Pol V Mut and RecA ATP hydrolysis rate constants

| | ATPase $k_{cat}$ (s⁻¹)* |
|---|---|
| pol V Mut WT | |
| no DNA | $(1.5 \pm 0.1) \times 10^{-3}$ |
| 12 nt oh HP | $(9.0 \pm 0.8) \times 10^{-3}$ |
| 3 nt oh HP | $(4.3 \pm 0.1) \times 10^{-3}$ |
| 30 nt ssDNA | $(160 \pm 5) \times 10^{-3}$ |
| RecA WT | |
| 30 ssDNA | $(100 \pm 5) \times 10^{-3}$ |
| pol V Mut E38K/K72R | |
| no DNA | $(1.7 \pm 0.2) \times 10^{-3}$ |
| 12 nt oh HP | $(4.4 \pm 0.7) \times 10^{-3}$ |
| 3 nt oh HP | $(3.4 \pm 0.7) \times 10^{-3}$ |
| 30 nt ssDNA | $(17 \pm 1) \times 10^{-3}$ |
| RecA E38K/K72R | |
| ssDNA | $(0.6 \pm 0.1) \times 10^{-3}$† |
| pol V Mut E38K/ΔC17 | |
| no DNA | $(7.0 \pm 1.5) \times 10^{-3}$ |
| 12 nt oh HP | $(54 \pm 9) \times 10^{-3}$ |
| 3 nt oh HP | $(46 \pm 2) \times 10^{-3}$ |
| 30 nt ssDNA | $(90 \pm 10) \times 10^{-3}$ |
| RecA E38K/ΔC17 | |
| 30 nt ssDNA | $(120 \pm 15) \times 10^{-3}$ |

*$k_{cat}$ is an average of at least three independent measurements; ± SEM.
†$k_{cat}$ was measured at 2.5 µM concentration for RecA E38K/K72R; ATP hydrolysis was not detectable at lower protein concentrations.

binds with higher affinity to the p/t DNA 12 nt oh (*Table 2*). DNA synthesis corresponds closely to ATPγS-dependent pol V Mut WT binding, showing a linear increase in primer extension up to about 180 µM ATPγS, reaching about 70% total primer usage at about 500 µM ATPγS (*Figure 3B*). There is no primer extension with ATP (*Figure 3B*). Pol V Mut E38K/K72R binds to p/t DNA with ATPγS (*Figure 3C*; *Table 2*). Although it does not appear to bind DNA appreciably in the presence of ATP in this assay, some binding must occur as DNA synthesis is clearly observed (*Figure 3D*).

The same experiments performed with pol V Mut E38K/ΔC17 show that binding and activity correlate well with ATPγS and ATP. Here, a much higher affinity to p/t DNA with ATP (*Table 2*) allows binding and primer extension (17% at 750 µM ATP, *Figure 3E,F*). Pol V Mut E38K/ΔC17 requires less ATPγS (78% at 100 µM ATPγS) for optimal binding and activity compared to the other pol V Muts, corresponding to the more robust activation of pol V consistently seen with this RecA variant (*Schlacher et al., 2006*; *Jiang et al., 2009*).

The data in *Figure 3* establish an absolute requirement for an ATP cofactor for pol V Mut binding to p/t DNA. In no case is DNA binding detected in the absence of ATP or an ATP homolog. Since binding is a precursor to synthesis, the same nucleotide requirement holds for pol V Mut-catalyzed primer extension (*Figures 1B*, *Figure 3*). The link that's missing is the role of ATP hydrolysis in the DNA binding/synthesis process. Indirect evidence hinting at just such a link is the observation that pol V Mut WT hydrolyzes ATP about 9-fold more rapidly on ssDNA than pol V Mut E38K/K72R and about 2-fold more rapidly than pol V Mut E38K/ΔC17 (*Table 1*), suggesting perhaps that the more rapid ATP hydrolysis by pol V Mut WT diminishes its ability to remain bound to DNA long enough to catalyze synthesis under these in vitro conditions. Pol V Mut WT-dependent primer extension significantly decreases with a mixture of ATP/ATPγS (*Figure 3—figure supplement 2*), further suggesting that the fraction of enzyme that binds and hydrolyzes ATP is not associated with DNA long enough to catalyze appreciable primer elongation. With p/t DNA, ATP hydrolysis is slower for pol V Mut E38K/K72R compared to pol V Mut WT (*Table 1*), which could account for incorporation with added ATP. In the case of pol V Mut E38K/ΔC17, although ATP hydrolysis is more rapid on p/t DNA compared to pol V Mut WT (*Table 1*), binding is much tighter (*Table 2*), which could plausibly explain primer extension observed with added ATP.

## ATP hydrolysis releases pol V Mut from p/t DNA

To obtain direct evidence for a possible role of ATP hydrolysis in the pol V Mut reaction pathway, we used pol V Mut formed with RecA E38K/ΔC17, which binds p/t DNA with higher affinity than either pol V Mut WT or pol V Mut E38K/K72R (measured $K_d$ for pol V Mut E38K/ΔC17 = 469 nM, compared to 876 nM for pol V Mut WT; *Table 2*). There is a concomitant ~5 to 10-fold greater DNA-dependent ATPase activity for pol V Mut E38K/ΔC17 compared with pol V Mut WT (*Table 1*). The pol V Mut E38K/ΔC17-p/t DNA complex formed in the presence of ATP was stable for a long enough time to conveniently monitor its dissociation (*Figure 4*). Fluorescent-labeled p/t DNA (12 nt oh, 50 nM) has a rotational anisotropy (RA) of 0.05 when diffusing freely in solution (*Figure 4*). When incubated in the presence of eightfold excess pol V Mut E38K/ΔC17 (400 nM), essentially all of the DNA is bound in a

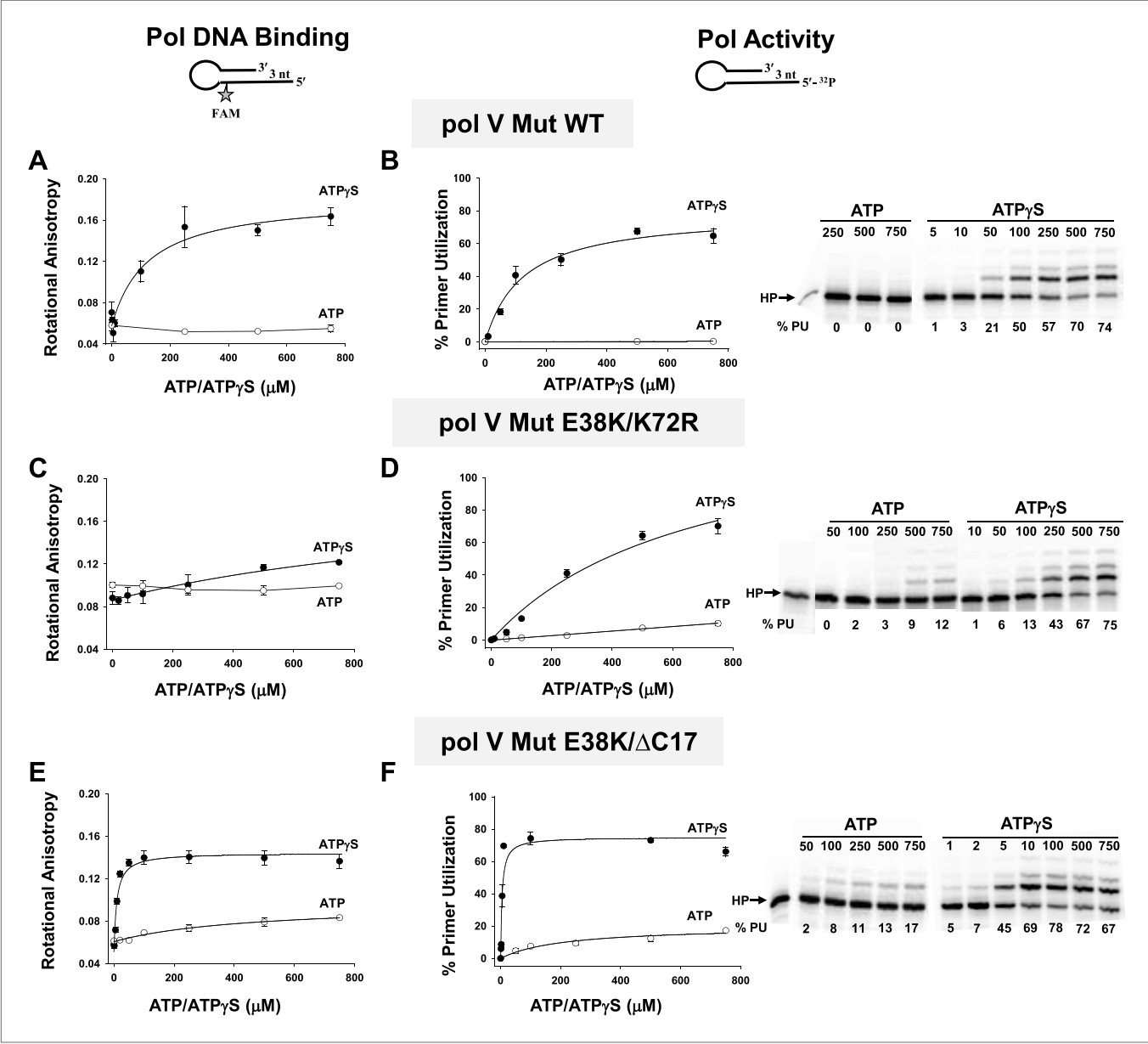

**Figure 3**. Pol V Mut binding and activity as a function of nucleotide. Binding of pol V Mut WT (1 µM) (**A**), pol V Mut E38K/K72R (400 nM) (**C**), and pol V Mut E38K/ΔC17 (400 nM) (**E**) to 3 nt oh HP was measured as a change in rotational anisotropy. Activity of pol V Mut WT (400 nM) (**B**), pol V Mut E38K/K72R (400 nM) (**D**), and pol V Mut E38K/ΔC17 (400 nM) (**F**) was quantified on 5'-$^{32}$P-labeled 3 nt oh HP with varying concentrations of nucleotide and 500 µM dNTPs. A gel showing primer utilization (%PU) as a function of nucleotide is presented to the right of the graph. ATP (open circle) and ATPγS (filled circle). The error bars correspond to 1 SEM.

The following figure supplements are available for figure 3:

**Figure supplement 1**. Pol V Mut binding to 12 nt oh HP DNA as a function of ATP/ATPγS.

**Figure supplement 2**. Pol V Mut WT activity as a function of ATPγS concentration in the presence of ATP.

complex with pol V Mut (RA = 0.12). Immediately following complex formation (t ~ 0), 160-fold excess unlabeled p/t DNA (8 µM) was added to trap pol V Mut as it dissociates. Pol V Mut E38K/ΔC17 forms a stable complex with p/t DNA in the presence of ATPγS, remaining bound for at least 4 min without dissociating. In contrast, when the complex is formed with ATP, pol V Mut E38K/ΔC17 dissociates exponentially as a function of time, asymptotically reaching ~100% free DNA at about 75 s (**Figure 4**).

**Table 2.** Pol V Mut affinity to 12 nt oh HP DNA

| Pol V Mut | $K_d$ (nM) | |
|---|---|---|
| | ATP | ATPγS |
| pol V Mut WT | nb | 876 ± 52 |
| pol V Mut E38K/K72R | nb | 920 ± 112 |
| polV Mut E38K/ΔC17 | 312 ± 46 | 469 ± 27 |

nb–binding not detected.

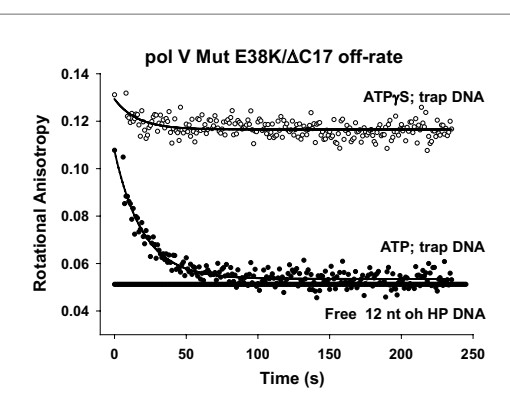

**Figure 4**. Pol V Mut E38K/ΔC17 off-rate in the presence of ATP and ATPγS. Fluorescence depolarization of fluorescein-labeled (12 nt oh HP) DNA was used to measure the dissociation constant of pol V Mut E38K/ΔC17 in the presence of ATP (filled circles) and ATPγS (open circles). A stable protein–DNA complex (400 nM and 50 nM, respectively) was preformed in the presence of nucleotide followed by the addition of excess (160 times) trap DNA (unlabeled 12 nt oh HP). The decrease in anisotropy over time was fit to an exponential decay to determine $k_{off}$ (0.053 ± 0.0025 s$^{-1}$).

The following figure supplements are available for figure 4:

**Figure supplement 1**. Pol V Mut E38K/ΔC17 primer extension length as a function of ATP or ATPγS concentration.

**Figure supplement 2**. Dissociation of pol V Mut WT and pol V Mut E38K/K72R in the presence of ATPγS.

**Figure supplement 3**. Pol V Mut remains intact in the presence of ATP/ATPγS and during DNA synthesis.

The off-rate determined as the first-order rate constant is 0.053 s$^{-1}$. The magnitude of the off-rate is in remarkably close agreement with the DNA-dependent ATPase single-turnover rate constant for pol V Mut E38K/ΔC17 = 0.054 s$^{-1}$ (*Table 1*). The data strongly suggest that ATP hydrolysis is responsible for pol V Mut-p/t DNA dissociation. This can also be seen during DNA synthesis, where longer segments of DNA are synthesized with ATPγS than with a similar concentration of ATP (*Figure 4—figure supplement 1*). The correspondence of dissociation rate to ATP hydrolysis rate further suggests that perhaps a single ATP turnover is sufficient to trigger dissociation of pol V Mut from a 3'-OH primer end. The absence of dissociation in the presence of the weakly hydrolyzed ATPγS strongly reinforces this conclusion. Pol V Mut WT and pol V Mut E38K/K72R, act similarly to pol V Mut E38K/ΔC17, remaining stably bound to p/t DNA in the presence of ATPγS (*Figure 4—figure supplement 2*). Although ATP hydrolysis affects pol V Mut–DNA complex stability, the integrity of the pol V Mut protein complex is not affected by hydrolysis, with RecA remaining bound to UmuD'$_2$C (*Figure 4—figure supplement 3A,B*).

The failure of pol V Mut WT to synthesize DNA in the presence of ATP can presumably be attributed to its inability to form a sufficiently stable complex with p/t DNA (*Figure 3A*). In contrast, pol V Mut E38K/ΔC17, which forms a much more stable complex with p/t DNA can synthesize DNA in the presence of either ATP or ATPγS (*Figure 3F*). To determine if pol V Mut WT can synthesize DNA in the presence of ATP (as must presumably happen in vivo) we have used the β sliding clamp to enhance pol V Mut binding to the 3'-primer end. Pol V Mut WT binds to β clamp and is able to synthesize DNA with moderately high processivity (*Karata et al., 2012*). To determine if increased binding stability for pol V Mut WT might enable it to use ATP for synthesis, we used a p/t DNA with a 12 nt oh-containing streptavidin attached at the 5'-template end and the hairpin loop, preventing the β clamp from sliding off the DNA (*Schlacher et al., 2006*). Pol V Mut WT in the presence of ATP (in the absence of ATPγS) incorporates 12 nt processively to reach the end of the template strand (*Figure 5*). The presence of ATP in the reaction, which is required to load β clamp (*Bertram et al., 2000*), is able to support DNA synthesis. However, the addition of ATPγS significantly stimulates synthesis (*Figure 5*), presumably by further enhancing binding to p/t DNA (*Figure 5*; *Table 2*), and by reducing the unloading of β clamp by the γ complex (*Bertram et al., 2000*).

## Discussion

Mutagenic DNA synthesis during the SOS response is an act of cellular desperation, and it comes with a price. However, when pol V Mut is activated and arrives on the scene, mutagenesis is not

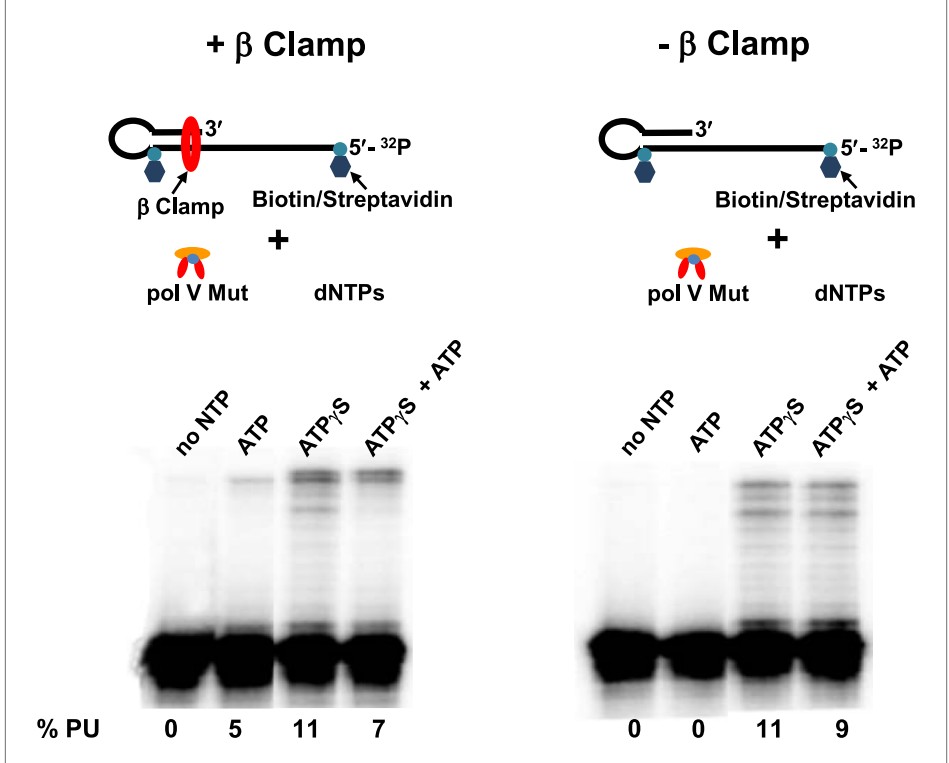

**Figure 5**. Pol V Mut WT is active with ATP only in the presence of β/γ complex. Sketch of the experimental set up is illustrated above the gels. To prevent the β clamp from sliding off the DNA, a 12 nt oh HP was designed containing biotin/streptavidin on both sides of the primer terminus substrate. The activity of pol V Mut WT was measured in the presence and absence of β/γ. In the presence of β/γ complex (left panel) pol V Mut WT is able to extend p/t with ATP, in contrast no DNA synthesis is observed with ATP in the absence of β/γ complex (right panel).

indiscriminate. Pol V Mut possesses a novel mechanism with which to limit processivity and restrict mutagenic DNA synthesis to those short DNA segments where it is required. In essence, the enzyme has evolved to do the absolute minimum to get cellular DNA synthesis restarted. No other DNA polymerase characterized to date has either an intrinsic ATPase activity or a similar autoregulatory mechanism.

We have previously shown that the active form of DNA polymerase V is UmuD'$_2$C-RecA-ATP (*Jiang et al., 2009*), but the roles of ATP and RecA in polymerase function were unknown. In vitro results using RecA variants that provide more facile activation and/or lack intrinsic ATPase function now elucidate the role of ATP. ATP or an ATP homologue is required for pol V Mut function (*Figure 1B*). Further, pol V Mut is a DNA-dependent ATPase (*Figure 2A,B*), which binds ATP in the absence of DNA (*Figure 2C*). Hydrolysis of ATP by pol V Mut results in dissociation of the enzyme from DNA (*Figure 4*). If ATPγS is used such that ATP is not hydrolyzed, then the enzyme remains stably bound to DNA (*Figure 4*, *Figure 4—figure supplement 2*). This is the only DNA polymerase studied to date that is regulated by ATP binding and hydrolysis.

Activation of pol V to pol V Mut requires transfer of a RecA monomer from the 3'-proximal tip of RecA* (*Schlacher et al., 2006*) (see e.g., *Figure 1A*). The ATP hydrolytic sites in RecA protein filament are situated at the interfaces of adjacent RecA subunits, and each neighboring subunit contributes key residues to the active site. There are no readily identifiable ATP-binding motifs present in either the UmuD' or UmuC subunits of pol V. The avid ATPase activity observed with the ATPase-deficient RecA E38K/K72R indicates that the RecA subunit is not contributing a Walker A motif (or P-loop) to the aggregate site. We speculate that the ATPase active site in pol V Mut is newly created when RecA is added to the complex during activation, perhaps at the interface between the RecA subunit and UmuC or UmuD'. This interface appears to play a central role in polymerase activation because the RecA(S117F) mutant (initially referred to as RecA1730), was shown to be SOS-non mutable in vivo

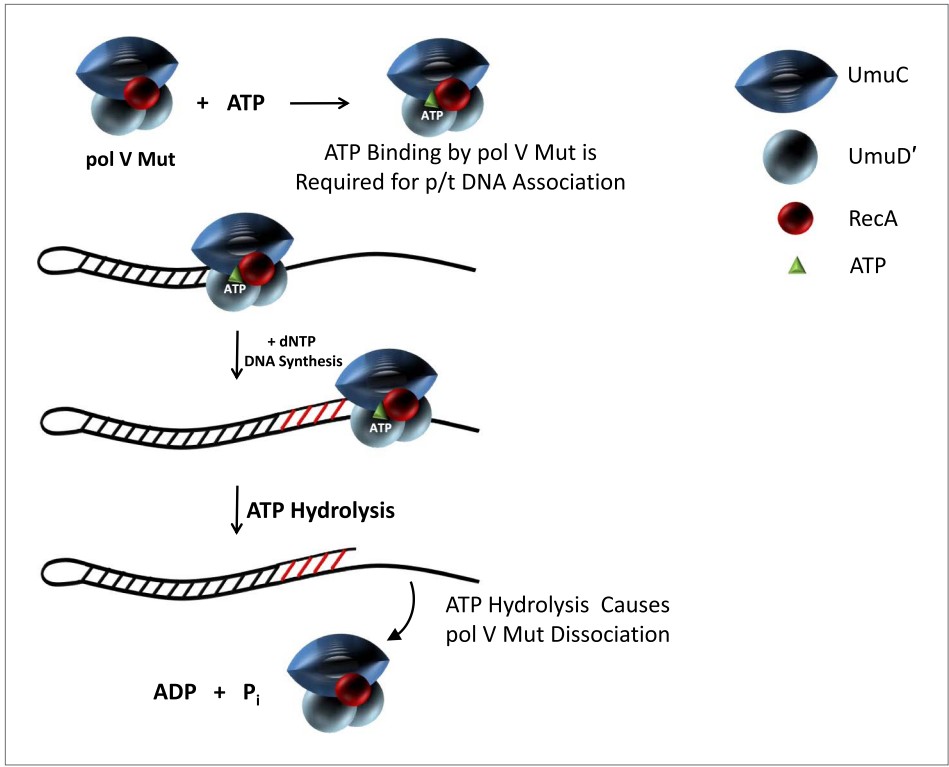

**Figure 6**. Model showing ATP regulation of pol V Mut activity. Pol V Mut is active for DNA synthesis only after binding a molecule of ATP (green triangle) to form UmuD'$_2$C-RecA-ATP. The binding of ATP is required for polymerase association with p/t DNA. ATP-hydrolysis catalyzed by an intrinsic DNA-dependent ATPase triggers pol V Mut-p/t DNA dissociation, while leaving intact the UmuD'$_2$C-RecA complex.

(*Dutreix et al., 1989*). The S117 residue is situated at the 3'-proximal tip of RecA* (*Sommer et al., 1998*), and, notably, pol V Mut S117F has no measurable DNA synthesis activity in vitro (*Schlacher et al., 2005*).

In the cell pol V is post-transcriptionally regulated through proteolysis (*Frank et al., 1993*; *Gonzalez et al., 1998*, *2000*) and by RecA*(*Burckhardt et al., 1988*; *Nohmi et al., 1988*; *Shinagawa et al., 1988*; *Schlacher et al., 2006*), presumably to ensure that this low fidelity pol (*Reuven et al., 1999*; *Tang et al., 1999*) is used only in dire circumstances. When the regulation of pol V fails, as it does for the constitutive RecA E38K mutant, pol V Mut is induced in the absence of DNA damage generating ~100-fold increase in mutations (*Witkin, 1967*). This huge increase in mutations likely occurs by copying undamaged DNA with exceptionally poor fidelity (*Tang et al., 2000*). The regulation of pol V is needed to limit mutations, especially in stationary phase cells (*Corzett et al., 2013*). Our biochemical data suggest that in vivo ATP adds another level to this regulation; not only is the timing of pol V activity regulated, but also its access to p/t DNA. A model for the role of ATP in pol V Mut activity is sketched in *Figure 6*. ATP is required to bind pol V Mut to DNA. ATP hydrolysis releases the enzyme from DNA, which in vivo would ensure that tracts of DNA synthesized by pol V Mut are short, limiting the opportunity for misincorporation to the region immediately adjacent to the template lesion. Therefore, the internally regulated DNA-dependent ATPase of pol V Mut provides a way to limit mutational load.

## Materials and methods

### DNA oligos
DNA oligos were synthesized using a 3400 DNA synthesizer (Applied Biosystems/Life Technologies, Carlsbad, CA). Oligo modifications (Flourescein-dT Phosphoramidite, Biotin-dT, and 5'-Amino-Modifier C12) were purchased from Glen Research. DNA sequences are in *Table 3*.

**Table 3.** Sequences for p/t HP DNA

| Sequences for p/t HP DNA | |
| --- | --- |
| 3 nt oh HP | 5′ AGA GCA GTT AGC GCA TTC AGC TCA TAC TGC TGA ATG CGC TAA CTG C 3′ |
| 3 nt oh HP (TTT) | 5′ TTT GCA GTT AGC GCA TTC AGC TCA TAC TGC TGA ATG CGC TAA CTG C 3′ |
| Fluorescein (**FAM**) 3 nt oh HP | 5′ AGA GCA GTT AGC GCA T(**FAM**)C AGC TCA TAC TGC TGA ATG CGC TAA CTG C 3′ |
| 12 nt oh HP | 5′ CGA AAC AGG AAA GCA GTT AGC GCA TTC AGC TCA TAC TGC TGA ATG CGC TAA CTG C 3′ |
| Fluorescein (**FAM**) 12 nt oh HP | 5′ CGA AAC AGG AAA GCA GTT AGC GCA TTC AGC TCA TAC TGC TGA A(**FAM**)G CGC TAA CTG C 3′ |
| Biotinylated (**Bio**) 12 nt oh HP | 5′ (**Bio**)GA AAC AGG AAA GCA GTT AGC GCA TTC AGC (**Bio**)CA TAC TGC TGA ATG CGC TAA CTG C 3′ |
| **Sequences for pol V Mut assembly, activity and ATP hydrolysis DNA** | |
| Amino C12-linked 45mer attached to cyanogen bromide-activated sepharose resin | 5′ **C12**TT TTT TTT TTT TTT TTT TTT TTT TTT TTT TTT TTT TTT TTT TTT T 3′ |
| ssDNA 30 mer for RecA transactivation and ATP hydrolysis | 5′ ACT GAC CCC GTT AAA ACT TAT TAC CAG TAA 3′ |

## Proteins

His-tagged pol V was purified from *E. coli* stain RW644 as described by *Karata et al. (2012)* and RecA WT was purified by a standard protocol (*Cox et al., 1981*). RecA E38K/K72R and RecA E38K/ΔC17 were provided by Michael Cox at the University of Wisconsin, Madison. We incorporated *p*-azido-L-phenylalanine (*p*AzF) (*Chin et al., 2002*) into RecA WT to site specifically label the protein with Alexa Fluor 488 DIBO alkyne. For the cloning of RecA, we used the pAIR79 plasmid, which was a gift from Michael Cox at the University of Wisconsin, Madison. The Phe21 sequence in RecA WT was replaced with the amber codon via site-directed mutagenesis using Pfu Ultra polymerase (Agilent Technologies). Once the sequence was confirmed, pAIR79 was cotransformed with the vector pEVOL-*p*AZF (a gift from the Peter Schultz lab at The Scripps Research Institute, San Diego, CA) into the BLR expression strains (*Young et al., 2010*). The RecA$_{F21AzF}$ protein was purified using the same standard protocol for RecA WT (*Cox et al., 1981*). RecA$_{F21AzF}$ was labeled with Alexa Flour 488 (RecA$_{F21AzF}$-Alexa Fluor 488) according to manufacturer's instructions (Life Technologies).

## Pol V Mut assembly

5′ amino-modified 45 nt ssDNA was covalently attached to Cyanogen-Bromide Sepharose resin according to manufacturer's protocol (Sigma–Aldrich) and transferred to a spin column (Biorad). Briefly, the resin was activated with 1 mM HCl then washed with water and equilibrated with coupling buffer (0.1 M NaHCO$_3$, 0.5 M NaCl). 5′amino-modified DNA (20 nmoles) was incubated with 100 mg resin overnight at 4°C, and unbound oligomers were removed by washing resin seven times with coupling buffer. Reactive amino groups were blocked with Ethanolamine (1 M, pH 8; Sigma–Aldrich) to prevent non-specific binding then stored in 1 M NaCl. The concentration of 45 nt ssDNA bound to the resin consistently provided about 6 nmoles per 100 mg of resin. Stable RecA* was assembled by incubating excess RecA or RecA mutants and ATPγS (Roche) or ATP (Amersham-Pharmacia) when stated with 0.5 nmole ssDNA-bound resin for 20 min at 37°C. Free RecA and ATPγS were separated from RecA*-resin by gentle centrifugation at 0.1×*g* for 1 min and collected in the flow through. Washes were repeated with reaction buffer (20 mM Tris–HCl pH 7.5, 25 mM Sodium Glutamate, 8 mM MgCl$_2$, 8 mM DTT, 4% glycerol, 0.1 mM EDTA) until no RecA was detected in the flow through. His-tagged pol V (2 nmole) was resuspended in reaction buffer and mixed with RecA*-resin to form pol V Mut. The pol V-RecA*-resin suspension was incubated at 37°C for 15 min followed by centrifugation (0.1×*g* for 1 min) to separate pol V Mut from the RecA*-resin in the spin column. The concentration of pol V Mut collected in the flow through was determined by SDS-PAGE gel. Briefly, pol V and RecA proteins at various known concentrations were resolved on an SDS-PAGE gel (10%) as standards and gel band intensities were quantified using IMAGEQUANT software. Standard gel intensities of UmuC and RecA were then used to determine the concentration of pol V Mut.

## DNA extension

The activity of pol V Mut was detected via DNA extension on a 5′-$^{32}$P-labeled primer template hairpin with a 3-nt overhang. Pol V Mut (400 nM) was added to a 10-μl reaction mixture containing annealed template DNA (50 nM), ATP, ATPγS, dATP, GTP, dTTP, or ADP (500 μM, unless stated otherwise) and

dNTPs (Amersham-Pharmacia) (500 μM each). Substrate DNA was preincubated with first streptavidin (400 nM) then β/γ complex (250 nM and 100 nM respectively) (a gift from Linda Bloom at the University of Florida, Gainesville) when present. Reactions were carried out at 37°C for 30 min. To detect free RecA in the pol V Mut solution, ssDNA (50 nM) was added to the reaction and activity was measured in the presence of ATPγS. Comparable activity levels between ATPγS alone and ATPγS + ssDNA indicate that pol V Mut is intact and free of RecA that is not part of the mutasome. Reactions were resolved on a 20% denaturing polyacrylamide gel allowing single nucleotide separation. Gel band intensities were detected by phosphorimaging and quantified using IMAGEQUANT software. Primer utilization was calculated as the unextended primer intensity subtracted from the total DNA intensity giving the percentage of primer utilized.

## Pol V Mut and RecA-dependent $P_i$ release (ATP hydrolysis)

*E. coli* phosphate-binding protein (PBP) was purified and labeled with MDCC fluorophore (Life Technologies) according to the protocol from *Brune et al. (1998)*. Binding of $P_i$ (phosphate) by MDCC-PBP is rapid and tight (Kd ~ 0.1 μM) resulting in a large increase in fluorescence (*Brune et al., 1998*). The change in fluorescence of MDCC-PBP was detected in real-time using a QuantaMaster (QM-1) fluorometer (Photon Technology International). Wavelengths for excitation and emission of the MDCC were selected using monochromators with a 1-nm band pass width. Excitation and emission were set at 425 and 464 nm, respectively. A 65-μl aliquot of 5 μM MDCC-PBP was incubated with 0.05 units/ml PNPase, 100 μM 7-methylguanosine and various concentrations of pol V Mut or RecA. The PNPase and 7-methylguanosine were used to remove any traces of Pi in the reaction prior ATP hydrolysis. The time-based scan was initiated for about 1000 s. ATP hydrolysis was initiated by adding a 5-μl mixture of ATP and DNA to final concentrations of 500 μM and 1 μM, respectively and the measurements were taken at 1 point per sec resolution. The maximum rate ($V_{max}$) of $P_i$ release was derived from the linear slope of Pi release, and $k_{cat}$ was calculated by dividing $V_{max}$ by the enzyme concentration. Each measurement was repeated 2–3 times.

## Steady-state rotational anisotropy binding assay and pol V Mut off-rate

Pol V Mut binding to p/t DNA was measured by changes in steady-state fluorescence depolarization (rotational anisotropy). Reactions (70 μl) were carried out at 37°C in standard reaction buffer, and contained a fluorescein-labeled hairpin DNA (50 nM), 500 μM ATP or ATPγS and varied concentrations of pol V Mut. For ATP titration experiments, fluorescein-labeled hairpin DNA (50 nM) and pol V Mut were mixed together in reaction buffer and ATP or ATPγS was titered to a final concentration of 750 μM. Rotational anisotropy was measured using a QuantaMaster (QM-1) fluorometer (Photon Technology International) with a single emission channel. Samples were excited with vertically polarized light at 495 nm and both vertical and horizontal emission was monitored at 520 nm.

To measure the enzyme's off-rate, first pol V Mut E38K/ΔC17 was prebound to 50 nM fluorescein-labeled hairpin DNA with a 12 nt single-stranded overhang in the presence of 500 μM ATP or 500 μM ATPγS. The enzyme off-rate was measured by monitoring changes in rotational anisotropy after the addition of excess trap, unlabeled DNA (8 μM). The off-rate was fit to single exponential decay. Anisotropy for free DNA and pol V Mut E38K/ΔC17 in the absence of trap was measured in the same experiment.

To determine the integrity of pol V Mut during ATP hydrolysis and DNA synthesis we employed the use of site-specifically labeled RecA$_{F21AzF}$-Alexa Fluor 488 to assemble pol V Mut WT$_{F21AzF}$-Alexa Fluor 488. Pol V Mut WT$_{F21AzF}$-Alexa Fluor 488 (100 nM) was mixed with dNTPs (500 μM) to a final volume of 70 μl and the rotational anisotropy was measured. To the same cuvette, ATP or ATPγS (500 μM) was added for another measurement. To create DNA synthesis conditions, 12 nt oh HP (1 μM) was added to the cuvette and rotational anisotropy was measured at 2 min, 5 min, and 10 min. The rotational anisotropy of RecA$_{F21AzF}$-Alexa Fluor 488 (100 nM) was measured in parallel to pol V Mut WT$_{F21AzF}$-Alexa Fluor 488.

## Binding of pol V Mut to etheno-ATP

Pol V Mut binding to etheno-ATP (Life Technologies) was measured as a change in rotational anisotropy at three different etheno-ATP concentrations, 500 μM, 750 μM, and 1000 μM. In a 70-μl reaction, etheno-ATP was mixed in standard reaction buffer with pol V Mut, pol V or RecA. The concentration of protein used for measurements was 400 nM. Rotational anisotropy was measured using a QuantaMaster (QM-1) fluorometer. Samples were excited with vertically polarized light at 410 nm and both vertical and horizontal emissions were monitored at 425 nm.

## Acknowledgements

ALE and MJ contributed to this work equally. We thank Dr Linda B Bloom, U Florida, Gainesville, for a generous gift of purified β/γ complex proteins and Dr Peter G Schultz, Scripps La Jolla, for kindly providing us with the vector pEVOL-*p*AZF.

## Additional information

### Funding

| Funder | Grant reference number | Author |
|---|---|---|
| National Institutes of Health | ES012259 | Aysen L Erdem, Malgorzata Jaszczur, Jeffrey G Bertram, Myron F Goodman |
| National Institutes of Health | GM21422 | Aysen L Erdem, Malgorzata Jaszczur, Jeffrey G Bertram, Myron F Goodman |
| National Institutes of Health | GM32335 | Michael M Cox |
| NICHD/NIH Intramural Research Program | | Roger Woodgate |

The funders had no role in study design, data collection and interpretation, or the decision to submit the work for publication.

### Author contributions

ALE, MJ, JGB, Conception and design, Acquisition of data, Analysis and interpretation of data, Drafting or revising the article; RW, MFG, Conception and design, Analysis and interpretation of data, Drafting or revising the article; MMC, Analysis and interpretation of data, Drafting or revising the article, Contributed unpublished essential data or reagents

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
