## [Decision Letter]

Thank you for sending your work entitled “DNA polymerase V activity is autoregulated by a novel intrinsic DNA-dependent ATPase” for consideration at *eLife*. Your article has been favorably evaluated by a Senior editor and 3 reviewers, one of whom is a member of our Board of Reviewing Editors.

The following individuals responsible for the peer review of your submission have agreed to reveal their identity: Michael Botchan (Reviewing editor) and Michael O'Donnell (peer reviewer).

The Reviewing editor and the other reviewers discussed their comments before reaching this decision, and the Reviewing editor has assembled the following comments to help you prepare a revised submission.

This is an important study, providing interesting answers to questions raised by previous work from the collaborating authors’ labs; why does the enzyme require ATP and what does the ribonucleotide do for the synthesis. The data clearly show that the SOS activated DNA Pol V, containing a single RecA molecule as part of the holoenzyme, requires ATP binding for association with a primer/template substrate and that hydrolysis leads to dissociation. The results are clearly presented and the data do provide very strong support for the idea that ATP hydrolysis serves as a switch to prevent the mutation prone polymerase from staying on the chromosome for too long and creating too many unwarranted mutations. It leaves the readers with the thought that during evolution, the Pol V co-opted the RecA, piggy-backing on it to use the RecA ATPase properties for its (Pol V) own design. Thus, RecA binds ATP in order to bind DNA, and hydrolyzes ATP to eject from DNA – and Pol V does the same. It is really quite brilliant and makes an excellent story about evolution of an enzyme that adapts to obtain extended mechanical features over those of simpler polymerases. The story should be published in *eLife*. A couple of points should be addressed in a simple revision.

1) The work relies heavily on mutant forms of RecA, particularly a mutant in the Walker A domain of the protein. The activated Pol V assembled with the mutant is thought to have an interesting hybrid ATP hydrolysis site where one or another surface of the other proteins may provide a residue compensating for the loss of the charge in the RecA. Two related views on this point arise: this mutant situation works well for synthesis and actually better than wt.

a) Can one eliminate the thought that in the wt the substitute surface actually doesn't come into play or do the authors believe that allostery distorts the well-studied RecA Walker A region? This point speaks to how “distinct” the RecA activity truly is in the polymerase or casts a shadow on where the ATPase site really lies.

b) Considering that the ATPase properties of RecA as a monomer may not have been studied before, it is conceivable that the conservative K to R replacement becomes functional in the context of a monomeric RecA. Though not essential we suggest the following – either for this paper or for a discussion of future directions: reconstitute pol V Mut with a RecA mutant that can no longer bind ATP (e.g., less conservative mutant) and determine if it still binds etheno-ATP, and to use the same mutant in ATPase assays. It seems easy to try, and may help support the interesting evolutionary scenario of a polymerase capturing another enzyme and harnessing its ATP-DNA cycle for enhanced mechanical function.

2) The wt activated Pol V actually hydrolyses ATP so well that essentially no synthesis is detected unless a non-hydrolyzable nucleotide is in the mix. With the mutant RecA enzymes ATPγS actually stimulates synthesis. It would be reassuring to show that for the wt and with quantification, changing the ratios of ATPγ to ATP leads to anticipated changes in synthesis. Quantification here might help with the next point.

3) There is little indication of the reproducibility of the experiments presented. Not one data plot in any of the figures have error bars nor do some of the tables give standard deviations. We would like to see this fixed in the revisions – one assumes that quantification of gel results with numbers of fold differences might be possible in some key experiments or with plots as indicated above.

4) For directly testing the switch and release idea – would using a much longer tail allow for a measurement of lengths actually synthesized? We anticipate that with a very long template only very short oligos would be made with ATP but at higher levels of the analogue longer lengths would be measured.

5) The authors suggest that pol V Mut remains intact in the presence of ATP/ATPγS, even during DNA synthesis. This conclusion needs to be firmed up. The experiment presented (Figure 4—figure supplement 2) uses the 12 oh p/t and the rotational anisotropy remains constant for up to 10 minutes. But what are the kinetics of DNA synthesis? They are not presented on either the 3 oh or 12 oh template, only 30 minute incubations are shown. So, is there DNA synthesis occurring under the conditions and time period of the rotational anisotropy experiment?

6) The extent of DNA synthesis shown in Figure 5 seems very low. The figure is over-exposed (for the starting material) that one cannot determine whether the first one or two nucleotides are being polymerized. It is difficult to compare this figure with, say, Figure 1, where extension of the first nucleotide seems quite efficient. Similarly, the extent of DNA synthesis seems to be less than previously reported in Figure 5 of Karata et al. (referenced within). What is the extent of synthesis?

---

## [Author Response]

*1) The work relies heavily on mutant forms of RecA, particularly a mutant in the Walker A domain of the protein. The activated Pol V assembled with the mutant is thought to have an interesting hybrid ATP hydrolysis site where one or another surface of the other proteins may provide a residue compensating for the loss of the charge in the RecA. Two related views on this point arise: this mutant situation works well for synthesis and actually better than wt*.

*a) Can one eliminate the thought that in the wt the substitute surface actually doesn't come into play or do the authors believe that allostery distorts the well-studied RecA Walker A region? This point speaks to how “distinct” the RecA activity truly is in the polymerase or casts a shadow on where the ATPase site really lies*.

RecA* nucleoprotein filament formed with the “non-ATP hydrolyzable” RecA E38K/K72R mutant has no detectable DNA-dependent ATP hydrolysis activity at RecA concentrations (0.2 and 0.4 μM). The same concentrations with pol V Mut E38K/K72R show significant DNA-dependent ATP hydrolysis, at rates that increase with increasing protein concentration (Figure 2). The weak DNA-dependent ATP hydrolysis detected at 2.5 μM RecA E38K/K72R (Figure 2—figure supplement 1), is less than the rate of hydrolysis observed using a 25-fold lower concentration of pol V Mut E38K/K72R (0.1 μM, Figure 2).

The strong DNA-dependent ATP hydrolysis observed when RecA E38K/K72R is used to form pol V Mut suggests that a distinct ATP hydrolysis pocket is formed. The much higher rate of ATP hydrolysis for pol V Mut WT (Figure 2) compared to pol V Mut E38K/K72R (Figure 2) suggests that the K72 residue in RecA, while not crucial to hydrolysis in pol V Mut, may well be proximal to the ATPase active site, or at least close enough to cause a significant reduction in the hydrolysis rate. The K72 residue (changed to R in the mutant) is at the 3′-proximal end of the RecA filament, and thus one might expect this surface to be close to UmuD′_2_C in the activated complex. At present, we can only say that the ATPase active site in pol V Mut does not feature the RecA K72 residue in its normal role.

*b) Considering that the ATPase properties of RecA as a monomer may not have been studied before, it is conceivable that the conservative K to R replacement becomes functional in the context of a monomeric RecA. Though not essential we suggest the following – either for this paper or for a discussion of future directions: reconstitute pol V Mut with a RecA mutant that can no longer bind ATP (e.g., less conservative mutant) and determine if it still binds etheno-ATP, and to use the same mutant in ATPase assays. It seems easy to try, and may help support the interesting evolutionary scenario of a polymerase capturing another enzyme and harnessing its ATP-DNA cycle for enhanced mechanical function*.

It is not possible to make pol V Mut with RecA that does not bind ATP. A mutant unable to bind ATP will not form a RecA* filament, and the assembly of an active pol V Mut requires pol V to interact with the 3′-tip of the RecA* nucleoprotein filament, which is required to transfer a RecA monomer to UmuD′_2_C (Jiang et al. 2009. The active form of DNA polymerase V is UmuD′_2_C-RecA-ATP. Nature, 460:359-63). We note that RecA cannot hydrolyze ATP as a monomer. In a RecA filament, the RecA active site for ATP hydrolysis is a composite site that is formed at the interface of two RecA subunits, both of which contribute residues essential to active site function. Thus, a RecA filament can hydrolyze ATP, a monomer cannot.

*2) The wt activated Pol V actually hydrolyses ATP so well that essentially no synthesis is detected unless a non-hydrolyzable nucleotide is in the mix. With the mutant RecA enzymes ATPγS actually stimulates synthesis. It would be reassuring to show that for the wt and with quantification, changing the ratios of ATPγ to ATP leads to anticipated changes in synthesis. Quantification here might help with the next point*.

We have performed the experiment suggested by the referees by changing the ratio of ATPγS/ATP, and obtained the anticipated result that DNA synthesis measured with ATPγS is reduced significantly in the presence of ATP. We measured DNA synthesis with pol V Mut WT as a function of ATPγS concentration, in the presence and absence of ATP. ATP (500 μM) reduces primer elongation (Figure 3—figure supplement 2, revised manuscript), which suggests that the fraction of enzyme that binds and hydrolyzes ATP is not associated with DNA long enough to observe activity.

*3) There is little indication of the reproducibility of the experiments presented. Not one data plot in any of the figures have error bars nor do some of the tables give standard deviations. We would like to see this fixed in the revisions – one assumes that quantification of gel results with numbers of fold differences might be possible in some key experiments or with plots as indicated above*.

Error bars have been added to figures and standard errors of the mean have been included in the tables.

*4) For directly testing the switch and release idea – would using a much longer tail allow for a measurement of lengths actually synthesized? We anticipate that with a very long template only very short oligos would be made with ATP but at higher levels of the analogue longer lengths would be measured*.

We performed the suggested experiment, which is included as Figure 4—figure supplement 1 in the revised manuscript. We measured DNA synthesis on a 12 nt oh HP with pol V Mut E38K/ ΔC17 as a function of ATP and ATPγS concentration. As expected short segments of DNA are made with ATP and much longer segments are synthesized with ATPγS, and primer extension lengths are greater at higher ATPγS concentrations.

*5) The authors suggest that pol V Mut remains intact in the presence of ATP/ATPγS, even during DNA synthesis. This conclusion needs to be firmed up. The experiment presented (Figure 4—figure supplement 2) uses the 12 oh p/t and the rotational anisotropy remains constant for up to 10 minutes. But what are the kinetics of DNA synthesis? They are not presented on either the 3 oh or 12 oh template, only 30 minute incubations are shown. So, is there DNA synthesis occurring under the conditions and time period*
*of the rotational anisotropy experiment?*

We have measured the DNA synthesis rates under the conditions (100 nM pol V Mut and 1 μM HP) and time period of the rotational anisotropy experiment; the data are shown in Figure 4—figure supplement 3, in the revised manuscript. Pol V Mut activity data were obtained at 2, 5, 10, 20 and 30 min, and synthesis is observed at 5, 10 and 30 min (at this high (10:1) DNA/pol V Mut ratio used to measure rotational anisotropy of RecA bound to UmuD′_2_C in the pol V Mut complex, primer extension is below the level of detection at 2 min). Therefore, we observe DNA synthesis occurring under the conditions and time period of the rotational anisotropy measurement for pol V Mut RecA_F21AzF_ -Alexa Fluor 488, similar to synthesis observed with the non-fluorescently labeled pol V Mut RecA WT (Figure 4—figure supplement 3).

*6) The extent of DNA synthesis shown in Figure 5 seems very low. The figure is over-exposed (for the starting material) that one cannot determine whether the first one or two nucleotides are being polymerized. It is difficult to compare this figure with, say, Figure 1, where extension of the first nucleotide seems quite efficient. Similarly, the extent of DNA synthesis seems to be less than previously reported in Figure 5 of Karata et al. (referenced within). What is the*
*extent of synthesis?*

In the Karata et al. reference, a different DNA substrate was used (circular p/t DNA) that could account for an increased rate of DNA synthesis. We also suspect that there had been a small amount of adventitious RecA present in the earlier report that could assemble as RecA* on unannealed primer strands and increase synthesis by transactivation of pol V Mut. Our significantly improved method of assembling pol V Mut using ssDNA covalently attached to Sepharose beads (Figure 1) eliminates ATP/ATPγS (and free RecA). The purpose of this experiment (Figure 5) was to determine if activity for pol V Mut WT can be obtained with ATP in the presence of the β clamp. The overall activity in the presence of ATP is low as a result of the dynamic behavior of γ complex using ATP to load and unload β. Additionally in these experiments the HP p/t DNA has biotin/streptavidin to prevent the loaded β clamp from sliding off, which partially inhibits pol V Mut activity. The key point is that pol V Mut WT DNA synthesis occurs with ATP in the presence of β/γ complex.